

# The nematode *Caenorhabditis elegans* enhances tolerance to landfill leachate stress by increasing trehalose synthesis

Yuru Chen[1,*], Binsong Jin[1,*], Jie Yu[1], Liangwei Wu[1], Yingying Wang[2], Bin Tang[1] and Huili Chen[1]

[1] College of Life and Environmental Sciences, Hangzhou Normal University, Hangzhou, Zhejiang, China
[2] National Wetland Museum of China, Hangzhou, Zhejiang, China
[*] These authors contributed equally to this work.

## ABSTRACT

The burgeoning issue of landfill leachate, exacerbated by urbanization, necessitates evaluating its biological impact, traditionally overshadowed by physical and chemical assessments. This study harnesses *Caenorhabditis elegans*, a model organism, to elucidate the physiological toxicity of landfill leachate subjected to different treatment processes: nanofiltration reverse osmosis tail water (NFRO), membrane bioreactor (MBR), and raw leachate (RAW). Our investigation focuses on the modulation of sugar metabolism, particularly trehalose—a disaccharide serving dual functions as an energy source and an anti-adversity molecule in invertebrates. Upon exposure, *C. elegans* showcased a 60–70% reduction in glucose and glycogen levels alongside a significant trehalose increase, highlighting an adaptive response to environmental stress by augmenting trehalose synthesis. Notably, trehalose-related genes in the NFRO group were up-regulated, contrasting with the MBR and RAW groups, where trehalose synthesis genes outpaced decomposition genes by 20–30 times. These findings suggest that *C. elegans* predominantly counters landfill leachate-induced stress through trehalose accumulation. This research not only provides insights into the differential impact of leachate treatment methods on *C. elegans* but also proposes a molecular framework for assessing the environmental repercussions of landfill leachate, contributing to the development of novel strategies for pollution mitigation and environmental preservation.

## INTRODUCTION

With the advancement of China's urbanization process, urban garbage is growing at an average rate of 8–10% and is expected to increase to a total 409 million tons by 2030 (*Wu et al., 2014*). Restricted by economy and technology, sanitary landfill is the main way to treat urban garbage (*Dhamsaniya et al., 2023*), yet secondary pollution caused by landfill leachates cannot be ignored. Landfill leachate refers to high-concentration effluents of organic wastewater resulting from external inflow as water passes through the garbage and overburden layers, liberating solutes such as ammonia nitrogen, heavy

Corresponding authors
Bin Tang, tbzm611@hotmail.com
Huili Chen, huilichen@hznu.edu.cn

metals, organic substances, and other substances with complex chemical composition. These discharges invoke large changes in local water quality and toxicant-load quantity (*Costa, Greice & Raquel Campos, 2019*; *Dai et al., 2011*; *Miao et al., 2019*; *Nie, 2000*; *Yuan et al., 2020*). Because of these characteristics, landfill leachate is a huge potential ecological threat, adversely impacting human health (*Laiju, Gandhimathi & Nidheesh, 2023*). So, the treatment process and toxicity evaluation of landfill leachates are hot and difficult points in the context of urbanization in the world. At present, membrane bioreactor treatment (MBR) and nanofiltration reverse osmosis (NFRO) are the most commonly used processes in the combined treatment process of landfill leachate (*Smailagi et al., 2020*).

The effect evaluation of various treatment processes for leachate is the basis for assessing safe discharge of leachate. The traditional assessment methods use chemical analysis that mainly identifies individual pollutants in the landfill leachate to evaluate the collective harm of the leachate, but these methods are often limited to physical and chemical characteristics (*Marttinen et al., 2003*; *Slack, Gronow & Voulvoulis, 2005*) that cannot reveal the complex interaction between pollutants and do not provide any rating of the biological effects of toxic substances. Compared with chemical analysis methods, biological analysis methods have more advantages in assessment of environmental risk. Biological methods have several advantages such as rapid-processing, high sensitivity, and cost-effectiveness (*Baderna, Caloni & Benfenati, 2018*; *Da Costa et al., 2018*), integrating the sum biological effects of all compound factors such as bioavailability, synergism, and target antagonism.

*Caenorhabditis elegans* is a soil nematode that has been widely used in developmental biology and cell biology as a standard model animal. *C. elegans* takes about three days from fertilization to adulthood, and its lifespan is often around two weeks. Because of its small size, transparent body, easy reproduction, short generation cycle, fully-characterized genetic background, sensitivity to interference, 60–80% homologous genes, and 12 metabolic signaling pathways shared with humans and many other characteristics (*Chen et al., 2019*; *Chen, Liu & Zhang, 2021*), *C. elegans* has also been increasingly used in the biological toxicity evaluation of hazardous substances. The evaluation end-point of acute toxicity, body bending, head swing, and other movement indicators, body length, the number of fertilized eggs is often used for evaluation of development and reproduction (*Wei et al., 2021*). However, although the toxicity of landfill leachate has been widely studied in luminescent bacteria, algae, plants, and mammals, only crustaceans have been studied in invertebrates. *C. elegans* has not been widely used in the toxicity assessment of landfill leachate (*Zhu et al., 2023*). Therefore, it is necessary for us to fill the research gap on the impact of landfill leachate on *C. elegans*, in order to provide theoretical basis for the impact of landfill leachate on humans.

Cells have various responses to environmental stress and transcriptional analyses indicate that cells can respond to extreme temperature and high permeability stresses by changing carbohydrate and glycerol metabolism (*Jiang et al., 2022*; *Łopieńska-Biernat et al., 2019a*; *Mollavali & Börnke, 2022*; *Wang et al., 2022*). Trehalose is a non-reducing sugar that exists widely in invertebrates such as nematodes (*Xu et al., 2021*) and insects (*Luo et al., 2022*; *Wang et al., 2021*; *Wang et al., 2020b*; *Yu et al., 2020*), functioning as one of the key molecules for both energy storage and as an effective anti-stress protective agent (*Luo et al.,*

*2022*; *Pellerone et al., 2003*; *Tang et al., 2018*; *Wang et al., 2020b*; *Wu, McAuliffe & O'Byrne, 2023*). Under non-stress conditions, trehalose is hydrolyzed into glucose by trehalase to provide energy for cells (*Elbein, 2004*; *Tang et al., 2018*). Under stress condition trehalose is synthesized extensively to protect biological cell activity. *Chen et al. (2018)* showed that the trehalose content in nematodes increased significantly under low temperature conditions to enhance its ability to protect cells. Under non-stress conditions, trehalose is hydrolyzed into glucose by *tre* to provide energy for cells (*Elbein, 2004*; *Shen et al., 2019*; *Tang et al., 2018*). Changes in trehalose content are closely related to trehalose-6-phosphate synthase and trehalase activities (*Chen, 2020*; *Yang et al., 2017*). Nematodes respond to high permeability environments by increasing the expression of trehalase gene during a specific period (0–30 min), but the related changes of trehalose metabolism of free-living nematodes in soil under leachate exposure. Its mechanism needs further study (*Wijnants, Vreys & Van Dijck, 2022*).

Here, we used *C. elegans* as a model to study of the effects of landfill leachate exposure on growth, development, and reproduction, in addition to assessment of trehalose content. By measuring life history parameters, clarify the treatment effects of various landfill leachate. The mRNA expression of trehalose related gene was detected by real-time fluorescence quantitative PCR as an indirect measure of trehalose metabolism to clarify the effect of leachate exposure on *C. elegans*. The present study examined the mechanism of an energy metabolism pathway of soil invertebrates in response to pollutant exposure and is intended to serve as a reference for toxicity evaluation and toxicity traceability of landfill leachate, also providing a basis for the improvement of landfill leachate treatment processes, further providing theoretical support for exploring the stress resistance mechanism of nematodes.

## MATERIALS AND METHODS

### Leachate collection

The raw leachate (RAW), membrane bioreactor (MBR) tail water, and nanofiltration reverse osmosis (NFRO) tail water treatments were collected from Shanghai Laogang landfill in March 2022 and stored at 4 °C. Use M9 buffer (0.6 g $KH_2PO_4$, 1.2 g $Na_2HPO_4$, 1 g NaCl, one mL 1M $MgSO_4$, $H_2O$ to 200 mL) (*Wei et al., 2012*) as the control group. The daily monitoring results of the landfill site show that the COD of the raw water leachate is about 7,000 mg/L, the pH is 7.94, the TN is 710 mg/L, and the $NH_4^+$-N is 684 mg/L. After MBR and NFRO treatment, the COD, pH, TN, and $NH_4^+$-N of the leachate respectively were 3,840 mg/L, 8.4, 510 mg/L, 16 mg/L and 264 mg/L, 7.62, 145 mg/L, and 0.6 mg/L (*Wang et al., 2020a*).

### Nematode culture

A wild-type *C. elegans* strain was obtained from the United States Caenorhabditis Genetics Center (CGC). The uracil-deficient *Escherichia coli* OP50 consumed by *C. elegans* was also acquired from CGC (*Brenner, 1974*).

Worms were collected at specific time point based on the normal life cycle of wild-type *C. elegans* raised at 20 °C. After 12 h of embryo hatching, the nematode reaches the L1 stage. At this point, the nematode body is transparent, with a body length of about 250 um

that can be observed under an anatomical microscope magnified by 5 times. The L3 stage nematode forms 28 h after embryo hatching, with a body length of 600 μm, that it is visible at two times magnification under a microscope, and organs such as the intestine can be observed. *C. elegans* were collected at 12 h (L1 larvae) and 28 h (L3 larvae) after hatching (*Kimble & Nüsslein-Volhard, 2022*). Nematode-specific NGM medium for culture of *C. elegans* was prepared as follows. Inoculate 1× 1 cm agar containing a large amount of L1 nematodes on NGM medium (1 g NaCl, 1 g cholesterol, 4 g Agar, $H_2O$ to 400 mL) coated with *E. coli* OP50 and incubate at a constant temperature at 20 °C. After 48 h of inoculation, M9 buffer was used to elute the nematodes on the medium into a 1 mL centrifuge tube. After centrifugation at 2,054 × g for 2 min, supernatant was removed and an alkaline cracking solution (1M NaOH: 5% NaClO = 1:1) was added (1 mL), followed by vortex mixing liquify 70% of the body of nematodes (*Zhang et al., 2022*). Centrifuge the lysate again and add 1 mL of K- medium (0.62 g NaCl, 0.48 g KCl, $H_2O$ to 200 mL) after removing the supernatant. After shaking and mixing, centrifuge again to remove the supernatant and wash nematode eggs, repeat this operation 2–3 times. Finally, separate the synchronized eggs of *C. elegans*. Inoculating synchronized nematode eggs onto the culture medium can obtain nematodes synchronized at different stages.

## Determination of survival rate and life history parameter

The relative survival rate within 72 h (*Zhu et al., 2023*) were measured by exposing L3 *C. elegans* to different treatment of landfill leachate (*Meyer & Williams, 2014*). Each treatment group has four replicates, with 30 nematodes per replicate. In order to exclude the influence of other factors, we use relative survival rate to represent. The calculation formula is as follows:

$$\text{Relative survival rate} = \frac{SN^{EX}}{SNE^{CK}} \times 100\%$$

$SN^{EX}$: the survival number of experimental group; $SNE^{CK}$: the mean of survival number of control group;

*C. elegans* L1 worms were exposed to different leachates for 24 h (acute exposure). There were 4 replicates in each group, with 30 nematodes in each replicate (*Kimble & Nüsslein-Volhard, 2022*). After the exposure, nematodes were washed 3× with K-medium, killed by water- bath heating (60 °C), and plated to slides for microscopic examination (Olympus, Tokyo, Japan). Body length, width of each nematode was processed with Image J software (Jena, Thuringia, Germany).

The measure of the brood number of *C. elegans* uses the L1 stage worms. A total of four treatment groups, each with 30 nematodes. After the exposure, nematodes were also killed by heating in a 60 °C water bath. Plated it under the microscopic examination, capture the picture and then measure the number of nematode eggs in the captured images.

Locomotion behavior reflected by body bending frequency and head swing frequency was used to indicate alteration in functional state of motor neurons (*Liu et al., 2022*). Behavioral assay using L1 larvae. After adding 1 ml K-medium to the NGM medium without the addition of *E. coli* OP50, each group picked 30 nematodes treated with different landfill leachate to different media. After a 1-minute recovery, *C. elegans* was transferred

to a slide containing K-medium following experimental treatments and observed by light microscopy. Record the number of head swings per minute and the number of body bends within 20 second of the nematode (*Xu et al., 2022*). Swinging the head of a nematode from one side to the other and then back indicates a head swing. The movement of the nematode relative to one wavelength in the long axis direction of the body is recorded as one body bend.

The detection of the numbers of fertilized eggs was carried out using L3 stage nematodes, with 30 nematodes per treatment group. After being exposed to different treatment groups for 24 h, each nematode was transferred to a new NGM medium, and the number of nematodes incubated in each medium within 48 h was observed as the number of fertilized eggs.

## RNA extraction and cDNA synthesis

About 10,000 *C. elegans* L3 worms were exposed to different leachates for 24 h and each group have four replicates. After 24 h, collect worms and extract total RNA.

Trizol kit (Invitrogen, Carlsbad, CA, US) was used to extract total RNA. Integrity of the total RNA extracted was determined by 1% agarose gel electrophoresis and RNA concentration and purity were determined with a NanoDrop™ 2000 spectrophotometer (Thermo Fisher Scientific, Waltham, MA, USA). Purified RNA was stored at −80 °C for subsequent experiments. PrimeScript® RT Reagent Kit with gDNA Eraser Kit (Takara, Kyoto, Japan) were used to synthesis of first strand complementary DNA (cDNA). Next, cDNA obtained was stored at −20 °C.

## Quantitative real-time polymerase chain reaction

Primer sequences used here were all obtained from GenBank accessions (Table 1). Primer Premier 5.0 software was used to verify functional primers (Premier Biosoft International, Palo Alto, CA, USA). The selected primers were synthesized by Zhejiang Shangya Biological Co. The specificity, concentration, and annealing temperatures of Primers were explored and the optimal amount and temperature of primers in the reaction were obtained by comparative analysis. The availability of primers was tested by agarose gel electrophoresis to verify whether the single size of the band was suitable to verify whether the primer produced non-specific amplification and primer dimer and the gel is stained with EB.

mRNAs were expressed by quantitative real-time polymerase chain reaction (qRT-PCR) using a SYBR Premix ExTaq™ fluorescent PCR kit, assayed with a Bio-rad CFX96™ Real-Time PCR Detection System (Bio-RAD Laboratories Inc., Hercules, CA, USA). Each PCR was performed in a final 10-μL volume, 1 μL cDNA, 0.4 μL (10 μM) of each primer, 3.2 μL ultrapure water, and 5 μL SYBR buffer. The reaction conditions were as follows: pre-denaturation at 95 °C for 5 s, denaturation at 95 °C for 30 s, and annealing and elongation at 59 °C for 30 s (35 cycles). Finally, the melting curve was drawn at 65−95 °C. Gene expression date were normalized to that of actin as the internal control (*Chen, 2020*). Data were analyzed by the $2^{-\Delta\Delta CT}$ method (*Livak & Schmittgen, 2001*) and using *actin* as an internal reference gene for correction.

**Table 1 Primers used for qRT-PCR.** *TPS*, trehalose-6-phosphate synthase gene. *TRE*, trehalase gene. *TPP*, trehalose-6-phosphate phosphatase synthase gene.

| Gene name | GenBank number | Forward primer (5′–3′) | Reverse primer (3′–5′) |
| --- | --- | --- | --- |
| QCetps-1 | NM_001047837.5 | CTTTGCAATTAACGCCGCAC | TATCGATCCAGACGAGAGTC |
| QCetps-2 | NM_064634.6 | ACTCACAGGGATCGTACAAA | ACCTCTTATAGGCCTGAAGC |
| QCetpp-1 | NM_078159.5 | TACGCTAGAGGAAATGAATGAC | ATGCCAAATGTGGTTCCTC |
| QCetre-1 | NM_059489.7 | ACTCAAAGRACCGAGACCAG | TGAGGGAACTGGACTATACAC |
| QCetre-2 | NM_068657.7 | ACTCAAAGRACCGAGACCAG | TGAGGGAACTGGACTATACAC |
| QCetre-3 | NM_001269432.3 | AGTGCTGCTGGAACAGAGCTT | ACCCATTTGGAAGCAATCAGG |
| QCetre-4 | NM_077848.9 | CACGCCAATTCACTTCCGATG | TGCCGATAGCTTGCAGACTTATC |
| QCetre-5 | NM_061248.5 | CACGCCAATTCACTTCCGATG | TGCCGATAGCTTGCAGACTTATC |
| QCeact-1 | NM_073418.9 | CTCTTGCCCATCAACCATG | CTTGCTTGGAGATCCACATC |

## Metabolism-related substance content

*C. elegan* L3 worms were exposed to different leachates for 24 h and each group have four replicates. After 24 h, collect worms and measure the cotent of carbonhydrate.

Trehalose, glucose and glycogen contents were detected with Glucose (GO) Assay kit (o-dinanisidine reagent, glucose oxidase/peroxidase reagent, glucose standard) (Hangzhou Jiecheng Biotechnology Co., Ltd., Hangzhou, China) and BCA protein determination kits (BCA reagent A, BCA reagent B, 5 mg/L Protein standards (BSA) (Hangzhou Jiecheng Biotechnology Co., Ltd., Hangzhou, China). In Briefly, the samples were homogenized in 200 $\mu$L phosphate buffered saline (PBS; pH 7.0), then added 800 $\mu$L PBS to the sample for ultrasonic crushing for 30 min. Then, the homogenate was centrifuged at 1,000 $\times$g for 20 min at 4 °C. Subsequently, 350 $\mu$L of the supernatant was removed and ultracentrifuged at 1,000 $\times$g for 20 min. Meanwhile, removed 350 $\mu$L of the supernatant to a new centrifuge tube and ultracentrifuged at 20,800 $\times$g for 60 min at 4 °C which was used to detect the contents of protein, trehalose and glycogen.

Trehalose content was detected by the anthrone method (*Leyva et al., 2008*; *Tatun, Singtripop & Sakurai, 2008a*; *Tatun et al., 2008b*) where 30 $\mu$L of 1% $H_2SO_4$ and 30 $\mu$L of 30% KOH was added to 30 $\mu$L of the sample, which was then placed in a 90 °C water bath for 10 min, and an ice bath for 3 min. Subsequently, 600 $\mu$L of developer (0.02 g of anthrone + 100 ml of 80% $H_2SO_4$) was added and the sample was placed in a 90 °C water bath for 10 min and cooled in an ice bath. The absorbance was then detected at 630 nm.

The content of glucose was detected by the kit (GO assay kit). Take 150 $\mu$L of the supernatant and suspension obtained by centrifugation at 20,800 $\times$g into different EP tube. Then, add 300 $\mu$L of glucose analysis reagent to the glucose standard curve. After a 30-minute water bath at 37 °C, add 300 $\mu$L sulfuric acid to terminate the reaction. Lastly, 12 N $H_2SO_4$ was added to terminate the reaction. Take 200 $\mu$L and divide it into three parallel point samples on an enzyme-linked immunosorbent assay (ELISA) plate. The absorbance value was detected at 540 nm. The detection methods of glycogen and glucose are similar, except that glucogen was aliquoted 160 $\mu$L and 32 $\mu$L of 0.1 U/L amylo-transglucosidase was added.

## Data analysis and statistics

The research uses Prism 9.0 statistical software package IBM SPSS Statistics 20.0 (IBM Corp., Armonk, NY, USA) to carry out. Before analysis, we used the Shapillo-Wilke method of SPSS software to test the normality of the data and used Levin's statistics to test whether the variance is uniform. After passing the test, statistical differences between different landfill leachates treated with different processes were tested using one-factor analysis of variance (one-way ANOVA) followed by Tukey's multiple comparison test. The effects of treatment group and exposure days on the relative survival rate of nematodes were analyzed using two-way ANOVA, and the Tukey's multiple comparison test was also used. For Tukey's new multiple range test, $P$ values 0.05 were considered to indicate significant differences, respectively. Figures were prepared using Prism 9.0 software. Data are represented as means $\pm$ standard error.

## RESULTS

### Effect of different landfill leachates on physiological toxicity of *Caenorhabditis elegans*

The relative survival rate (Fig. 1) of nematodes differed among landfill leachate ($F_{2,35} = 50.54$, $P < 0.001$), there are significant differences in relative survival rates due to differences in exposure days ($F_{2,35} = 14.250$, $P < 0.001$), with the untreated leachate group showing the highest worm mortality rate after short-term (1-day) exposure, with a relative survival rate of $62.0370 \pm 1.7730\%$, followed by membrane bioremediation (MBR) with a relative survival rate of $75 \pm 2.3302\%$. Compared with other treatment groups, the nanofiltration reverse osmosis tail water (NFRO) treatment group showed relatively stable nematode relative survival rate and lower toxicity over a longer exposure time.

Nematode body length ($F_{3,116} = 4.265$, $P = 0.0112 < 0.05$), body width ($F_{3,116} = 10.07$, $P < 0.0001$), head swing frequency ($F_{3,116} = 40.03$, $P < 0.001$), body bending frequency ($F_{3,116} = 62.35$, $P < 0.0001$), brood number ($F_{3,116} = 12.04$, $P < 0.001$) differed significantly among landfill leachate treatment groups (Fig. 2). Raw leachate (RAW) appeared to significantly inhibit the growth, behavior, and reproduction (Fig. 2) of nematodes, with the most stark contrasts observed for behavior. The head swing frequency and body bending frequency of nematodes exposed to raw leachate were only one tenth of those of control group (CK) exposed in M9 buffer (Figs. 2C, 2D). However, the inhibition of MBR group on nematode development was reduced. As shown in (Fig. 3), clear eggs can be seen in the CK and NFRO groups of *C. elegans*, but not in the MBR and RAW groups. The NFRO group showed significantly reduced toxicity and there were no significant differences between NFRO group and CK group in brood number, body width, body length and behavior.

### Changes in major carbohydrates of *Caenorhabditis elegans* treated with different landfill leachates

Compared with CK group, glucose and glycogen contents in both RAW and MBR groups decreased, and there is a significant difference between RAW and MBR groups (Figs. 4B, 4C). Trehalose content in RAW and MBR groups increased significantly after treated (Fig. 4A). The glucose and glycogen contents decreased about 70% in both RAW and MBR

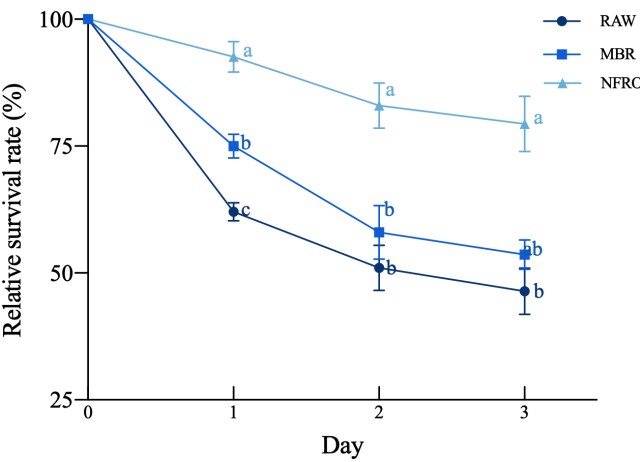

**Figure 1** **Effects of different landfill leachates on the survival of *Caenorhabditis elegans*.** RAW: raw leachate, MBR: membrane bioreactor, NFRO: nanofiltration reverse osmosis. Different lowercase letters (a, b and c) indicate significant differences at each observation time ($P < 0.05$).

groups. Meanwhile, the content of glucose, glycogen and trehalose of nematodes in NFRO group are similar to CK group (Fig. 4).

### Expression of genes related to trehalose metabolic pathway in *Caenorhabditis elegan s* treated with different leachates

mRNA expression of *tps-1, tps-2, tpp* (*gob-1*) in RAW group were significantly higher than those in the CK group, while the expression of *tre-4* showed no significant changes (Fig. 5). However, expression of *tre-1, tre-2, tre-3* increased to different degrees. Overall, changes in nematode gene expression favored trehalose synthesis. Furthermore, expression of *tps-1, tps-2, gob-1* in MBR group was similar to RAW group. In NFRO group, trehalose decomposition and synthesis genes of nematodes remained highly expressed and sugar content remained stable. This also accords with our earlier observation which showed that *C. elegans* mainly resist adversity by increasing the expression of the trehalose synthesizing genes.

## DISCUSSION

### The effect of landfill leachate on nematodes from the perspective of characteristics

Under the infiltration of landfill leachate treated with different processes, the nematodes exhibit different physiological states and trehalose related metabolic changes. Due to the characteristics of landfill leachate, its inhibitory effect on nematodes is most obvious, for the following reasons (Fig. 1). At the first, previous studies have shown that landfill leachate is a high concentration organic wastewater containing pollutants with high environmental longevity and producing a large water quality change. Therefore, its anoxic (*Wu et al., 2014*) and high osmotic pressure environment significantly inhibits the growth of nematodes. Secondly, the concentration of ammonia nitrogen in landfill leachates is extremely high.

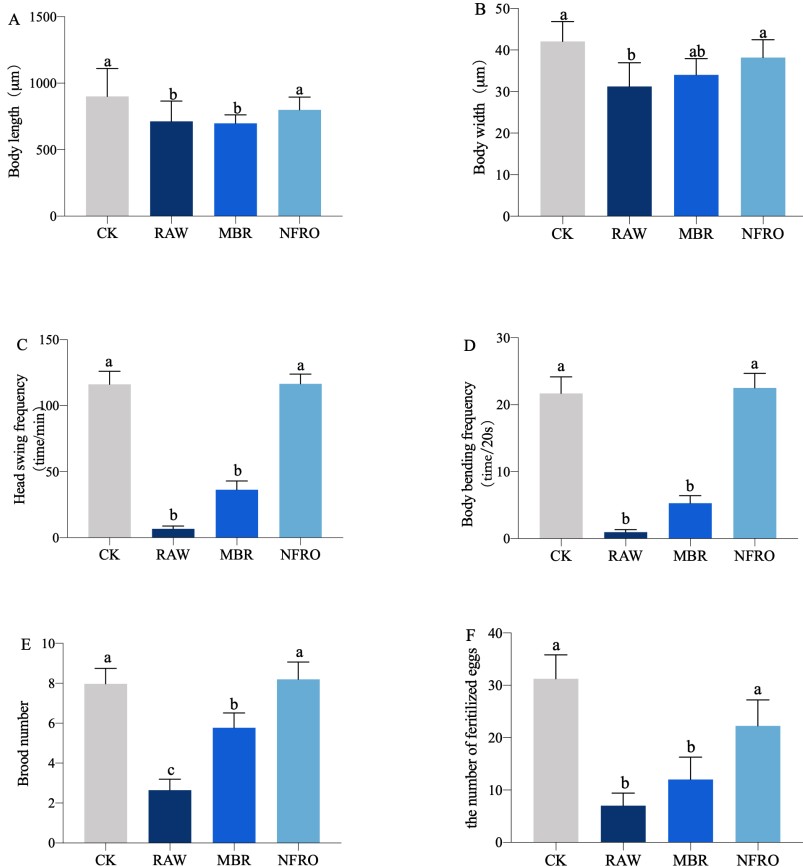

**Figure 2   Growth and reproduction characteristics of the nematode *Caenorhabditis elegans* after 24 h exposure to different landfill leachates.** (A) Body length; (B) body width; (C) head swing frequency; (D) body bending frequency; (E) brood number; (F) the number of fertilize eggs. CK, control group (M9 buffer); RAW, raw leachate, MBR: membrane bioreactor; NFRO, nanofiltration reverse osmosis. Different letters (a, b) indicate significant differences among treatments ($P < 0.05$).

When the concentration of ammonium in the soil increases, the resulting ammonium toxicity produces strong adverse effects on nematodes (*Wei et al., 2012*). In addition, the increase of soil ammonia nitrogen concentration directly leads to soil acidification, leading to loss of soluble cations and decrease in soil pH. The resulting reduction of available cations further increases soil acidification, which is not conducive to the survival of soil nematodes (*Li, He & Wu, 2014*). Third, landfill leachate is rich in toxic heavy metals (lead, cadmium, and nickel) (*Aydi et al., 2020*). A number of heavy metal pollution assessment studies on mining plants have shown that heavy metals can have a direct toxic impact on *C. elegans*, with cadmium and mercury levels negatively affecting their growth, survival, and reproduction (*Kang et al., 2023*; *Martinez et al., 2018*), consistent with our findings. Previous studies have also shown that inorganic mercury and methylmercury can inhibit the locomotory behavior, growth, and reproductive ability of worms (*McElwee & Freedman, 2011*), while heavy metals often combine to produce complex toxicity effects (*Moyson et al., 2018*; *Tang et al., 2019*).

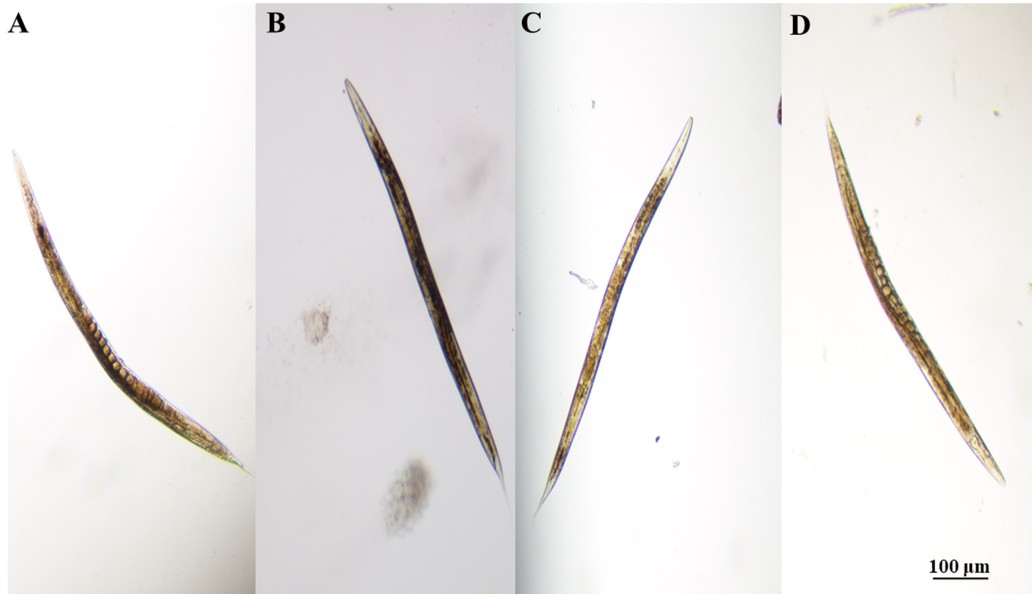

Figure 3 **Nematode egg carrying capacity after different exposures.** (A) CK, control group; (B) RAW, raw leachate; (C) MBR, membrane bioreactor; (D) NFRO, nanofiltration reverse osmosis.

## Change of the trehalose in different adversities

Trehalose is found throughout almost the entire life cycle of nematodes and is the blood sugar of nematodes (*Łopieńska-Biernat, Zaobidna & Dmitryjuk, 2015*; *Pellerone et al., 2003*). The decrease in trehalose content will not affect the survival of nematodes, but the trehalose content of nematodes will increase to adapt to adverse conditions such as low temperature and drought (*Behm, 1997*; *Liu et al., 2019*; *Łopieńska-Biernat et al., 2019a*; *Pellerone et al., 2003*). Previous studies have lacked research on the sugar metabolism of complex, high concentration organic wasteful water such as landfill leachates in biological systems. Therefore, we will explain the experimental results based on previous research and the characteristics of landfill leachate. Under the exposure of landfill leachate, the trehalose content of nematode increased due to its high osmotic pressure, low pH characteristics, and heavy metals and other toxic substances. Other studies on trehalose have shown that organisms can respond to acid and high osmotic pressure through increased trehalose content (*Chen et al., 2020*; *Gong et al., 2022*). *Erkut et al. (2019)* showed that the trehalose content of nematodes increased several times under extreme dehydration conditions, which is similar to our research results (Fig. 4). In addition, *Crowe, Crowe & Chapman (1984)* find trehalose plays a great role in dehydration conditions. Furthermore, compared with the control group (CK), nematode glycogen and glucose content in the landfill leachate treatment group (RAW) and the membrane bioreactor (MBR) treatment group significantly decreased, while the trehalose content increased (Fig. 4). Another study documented that glucose content in *Aedes albopictus* exposed to heavy metal cadmium similarly decreases (*Yu et al., 2020*).

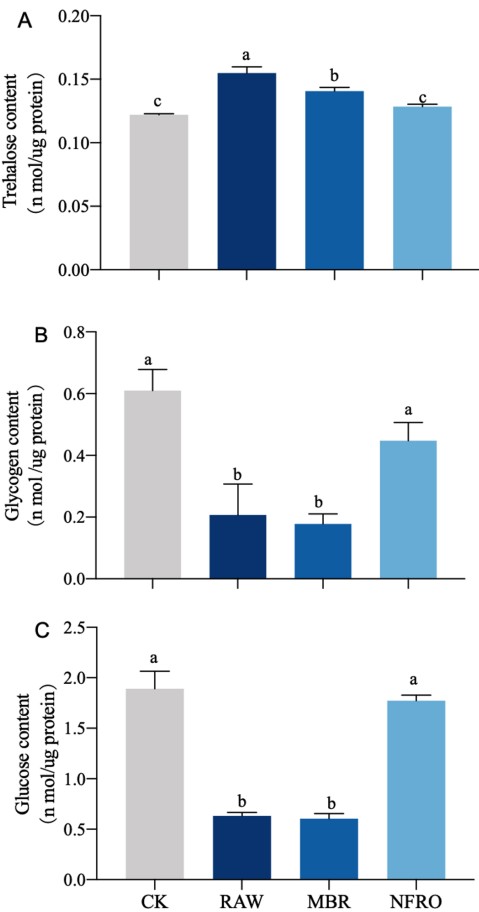

**Figure 4  Carbohydrates of *Caenorhabditis elegans* treated with different landfill leachates.** (A) Trehalose content; (B) glycogen content;(C) glucose content. CK, control group; RAW, raw leachate; MBR, membrane bioreactor; NFRO, nanofiltration reverse osmosis. Different letters (a, b) indicate significant differences among treatments ($P < 0.05$).

Our research also shows that the various life history parameters of nematode exposed to the waste leachate tail water treated by nanofiltration reverse osmosis (NFRO) are close to those of the control group (Fig. 2). Nanofiltration and reverse osmosis technology intercepts metals, minerals, and other substances with a diameter of more than 0.1 nm. This method can intercept up to 98.5% of COD, greatly reducing toxicity (*Zhu et al., 2022*). The membrane bioreactor (MBR) process tail water treatment method mainly plays a nitrification role as indicated by ammoniacal nitrogen measurements ($NH_3$-N) and the tail water often contains a large amount of refractory organic compounds (*Yang et al., 2022a*; *Yang et al., 2022b*). Furthermore, most commercial polymeric membrane material used in the membrane bioreactor (MBR) system has a high pollution potential, such as polyethersulfone (PES) and polyvinylidene fluoride (PVDF) (*Lemos et al., 2021*). The changes in the characteristics of the two types of landfill leachate treated with different processes not only alter the physiological effects of nematodes, but also alter their trehalose, glucose, and glycogen contents. In the MBR group with significant physiological effects,

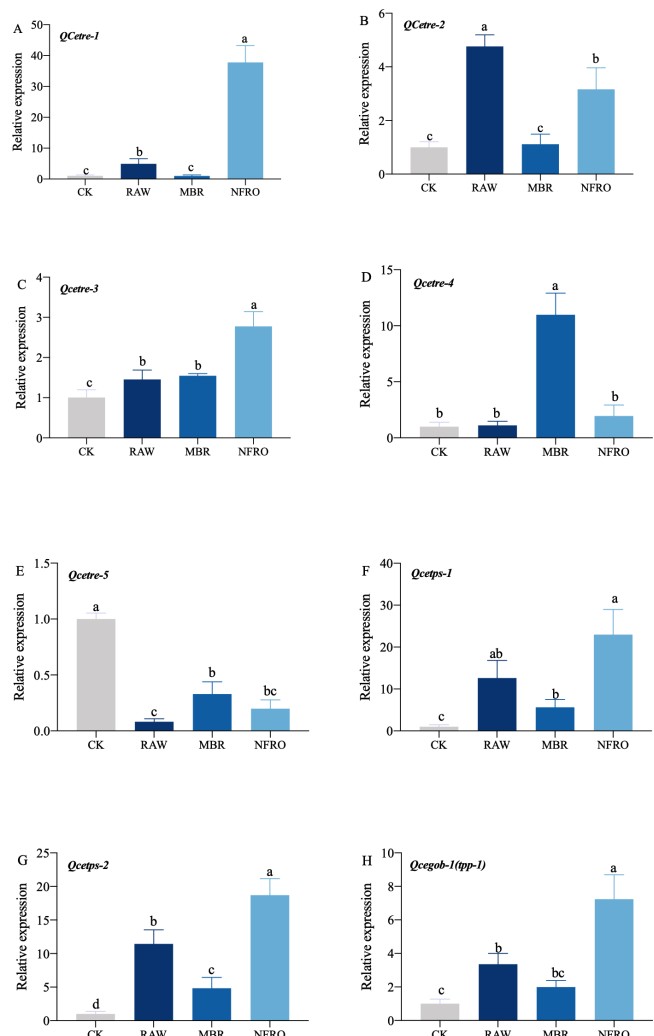

**Figure 5** Expression of genes related to trehalose metabolic pathway in *Caenorhabditis elegans* treated with different landfill leachates. (A) *QCetre-1*; (B) *QCetre-2*; (C) *QCetre-3*; (D) *QCetre-4*; (E) *QCetre-5*; (F) *QCetps-1*; (G) *QCetps-2*; (H) *QCegob-1 (tpp)*. CK, control group; RAW, raw leachate; MBR, membrane bioreactor; NFRO, nanofiltration reverse osmosis. Different letters (a, b) indicate significant differences among treatments ($P < 0.05$).

the changes in trehalose and other sugar contents were similar to those in the RAW group, while the NFRO group was similar to the CK group (*Wang et al., 2020a*).

## The accumulation of trehalose is mainly through synthesis

Under stress, trehalose in nematodes is a continuous synthesis and decomposition process (*Chen et al., 2018*). Following exposure to the tail water treatments examined here, the relative expression of *C. elegans* genes related to trehalose decomposition and synthesis in the nematode increased. Accumulation of trehalose in nematodes mainly depends on the increased expression of trehalose synthesis-related genes under stress conditions. Many studies have shown that the role of trehalose synthesis genes plays a more important role

in the survival of nematodes, and the silencing of trehalose synthesis genes is fatal to nematodes (*Elbein, 2004*; *Hespeels et al., 2015*; *Kushwaha et al., 2012*). Our experimental results also confirm this conclusion. The most significant increase in the expression of genes related to trehalose synthesis in nematodes treated with RAW and MBR indicates that nematodes overcome stress by upregulating the expression of *tps-1*, *tps-2*, and *gob-1* genes to increase trehalose synthesis (Fig. 5). Meanwhile, the study by *Łopieńska-Biernat et al. (2019b)* suggests that the synthesis of trehalose under stress conditions often comes at the cost of glycogen, which explains why the trehalose content in the midgut increases while the glycogen content decreases in the MBR and RAW treatment groups. Many studies on the mechanism of trehalose resistance of nematodes under stress have shown that under low temperatures, the trehalase gene *tre* expression of nematodes decreases, the trehalose synthesis-related genes *tps* and *tpp* increase, and protective trehalose accumulation occurs. Cold tolerance is directly proportional to the trehalose content in nematodes (*Chen, 2020*; *Chen et al., 2018*; *Dai et al., 2011*; *Zheng et al., 2022*), yet occurs under drought and high osmotic pressure conditions. Whether the trehalose content in nematodes increases is not absolute, but depends on exposure to adverse factors (*Qian, 2014*). The results of *Chen et al. (2016)* hypertonic stress experiments on rice stem tip nematode showed that the *ab-tre-1* gene expression of the nematode significantly increased within a short period of time after stress, and then gradually decreased, thereby improving its tolerance to stress. It indicates that under high permeability, trehalose first be hydrolyzed into glucose to maintain the osmotic balance of the internal environment, and then rapidly synthesizes stable membrane and protein structure. In the present study, after exposure to landfill leachates, the trehalose hydrolyzing gene *tre* of the nematode increased in a short period of time and change in expression was more significant specifically in *tre1*, *tre2*, and *tre3*, which is consistent with previous reports.

Currently, there is no in-depth study on the resistance trehalose-dependent mechanism of nematodes under toxic stress, but *Yu, Zhang & Sang (2022)* found that the metabolism of earthworm digestive and immune system-related metabolites (trehalose-6-phosphate, phosphate) in earthworms are upregulated under copper exposure, which indicates that the synthesis of trehalose-6-phosphate is closely related to *tps*. Our results indicate that under the treatment of landfill leachate (RAW) and nanofiltration reverse osmosis (NFRO), the *tps* gene significantly increased, but the trehalose-6-phosphate produced by nematodes in the NFRO treatment group was rapidly synthesized by *tpp*. Conversely, the trehalose-6-phosphate content in the RAW treatment group increased to adapt to highly toxic conditions. Interestingly, tail water as an organic solution with high osmotic pressure, high heavy metal content, and high ammonia nitrogen concentration produced a response in nematodes to stress is similar to that of high osmotic pressure responses. According to existing studies on trehalose genes in insects (*Wu et al., 2022*), it can be concluded that trehalose genes in nematodes play unequal roles in the process of stress resistance. *tps-1*, *tps-2*, *gob-1* plays a more important role in stress resistance (Fig. 6). Therefore, by detecting the content and gene expression of trehalose, glycogen, and glucose, it is possible to quickly assess the toxicity of landfill leachate and improve treatment technology.

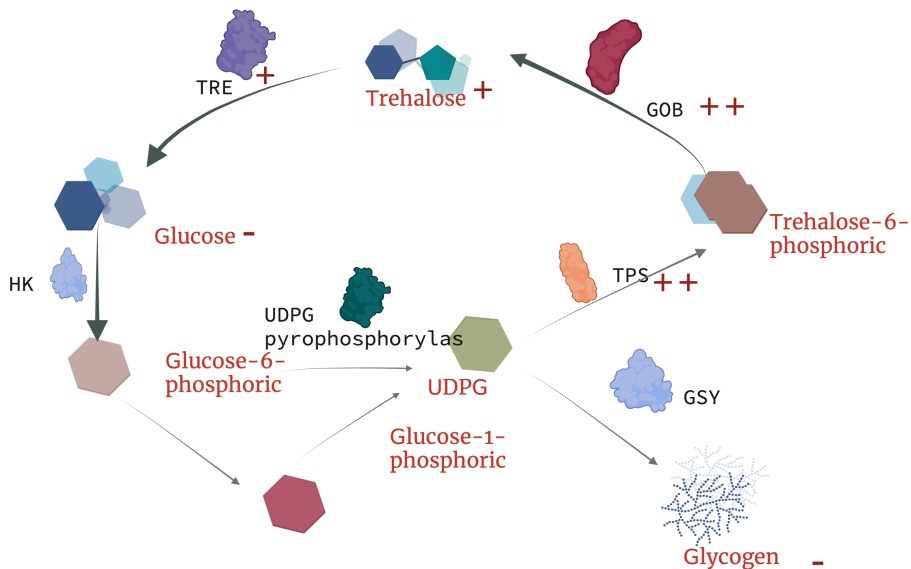

**Figure 6   Schematic diagram of trehalose response mechanism to landfill leachate stress in *Caenorhabditis elegans*.** TRE, trehalase; TPP(GOB), trehalose phosphate synthase; TPS, trehalose-6-phosphate synthase; HK, hexokinase; UDPG, uridine diphosphate glucose.

## The future and outlook

Our investigation delves into the alterations in trehalose metabolism in *C. elegans* when exposed to the complex organic matter present in landfill leachate, laying a theoretical groundwork for understanding the broader implications of landfill leachate on metabolic processes not only in nematodes but potentially in humans as well. Recognizing the varied sensitivities of different test species—including microorganisms, plants, and other invertebrates—to pollutants highlights the importance of future studies broadening the scope of monitoring. There is a promising avenue for establishing a comprehensive, multi-trophic level assessment system. This system would aim to characterize the environmental toxicity of leachates and other pollutants more holistically, across communities and ecosystems.

The comparative analysis of the membrane bioreactor (MBR) and nanofiltration reverse osmosis (NFRO) treatment processes within this study offers a preliminary foundation for refining the separation of organic matter in wastewater treatment strategies. Moreover, advancing our research to establish correlations between physiological indicators and the reproductive-related trehalose metabolism pathways could shed light on the underlying mechanisms of pollutant toxicity and pathogenicity. It opens the door to employing genetic engineering approaches to modulate the trehalose metabolism pathway in nematodes. Such innovations could pave the way for leveraging nematodes in bioremediation efforts, aiming to rehabilitate the soil and aquatic environments surrounding landfill sites. By focusing on these prospects, our research not only contributes to the fundamental understanding of stress response mechanisms in nematodes but also underscores the potential of this knowledge in environmental management and remediation technologies.

## CONCLUSIONS

Our research shows that landfill leachate has a significant negative impact on the survival, growth, and reproduction of nematodes. The toxicity of the effluent from the membrane biological method has been reduced, and there is almost no significant difference between the effluent from the nanofiltration reverse osmosis method and the control group. RAW and MBR tail water exposure also increased the trehalose synthesis and decreased the content of glycogen and glucose in *C. elegans*, while the expression of *tpp* (*gob*) and *tps* was significantly upregulated (Fig. 6). In summary, trehalose plays an important role in the response to nematode stress. Although the research on trehalose metabolism in insects in extreme environments has been extensive, current research rarely examines the effects on nematodes and actions of complex toxic compounds. However, in this study, we only preliminarily reveal the impact of the complex mixture of landfill leachate on trehalose metabolism. Therefore, future research will build a network that fully considers the characteristics of various toxic substances and other substances in landfill leachate to analyze its impact on trehalose metabolism mechanisms.

## ACKNOWLEDGEMENTS

We thank Zihao Gu for aid with the *Caenorhabditis elegans* culture. We thank Xueli Zhu with the help for providing landfill leachate. We thank LetPub for linguistic assistance and pre-submission expert review.

### Funding

This work was supported by the Basic and Commonweal Programme of Zhejiang Province (LGN21C140009), the Students' Innovation and Entrepreneurship Training Project of China (202310346056), and the Program of Potential Talents in Zhejiang Province (2022R426A016). The funders had no role in study design, data collection and analysis, decision to publish, or preparation of the manuscript.

### Grant Disclosures

The following grant information was disclosed by the authors:
Basic and Commonweal Programme of Zhejiang Province: LGN21C140009.
Students' Innovation and Entrepreneurship Training Project of China: 202310346056.
Program of Potential Talents in Zhejiang Province: 2022R426A016.

### Competing Interests

The authors declare there are no competing interests.

### Author Contributions

- Yuru Chen performed the experiments, analyzed the data, prepared figures and/or tables, and approved the final draft.

- Binsong Jin performed the experiments, analyzed the data, prepared figures and/or tables, authored or reviewed drafts of the article, and approved the final draft.
- Jie Yu performed the experiments, prepared figures and/or tables, and approved the final draft.
- Liangwei Wu performed the experiments, prepared figures and/or tables, and approved the final draft.
- Yingying Wang performed the experiments, analyzed the data, prepared figures and/or tables, and approved the final draft.
- Bin Tang conceived and designed the experiments, performed the experiments, authored or reviewed drafts of the article, and approved the final draft.
- Huili Chen conceived and designed the experiments, performed the experiments, authored or reviewed drafts of the article, and approved the final draft.

## Data Availability

Data is available at GitHub (https://github.com/binsong/Data_Leachate_Tolerance_in_C_elegans.git).

## Supplemental Information

Supplemental information for this article can be found online at http://dx.doi.org/10.7717/peerj.17332#supplemental-information.

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
