# Peer review of "The nematode Caenorhabditis elegans enhances tolerance to landfill leachate stress by increasing trehalose synthesis"

_PeerJ, doi:10.7717/peerj.17332_

## Round 0.1 · original submission · Minor Revisions

Thank you for submitting your work to PeerJ. Please submit a new version addressing the reviewer comments.

Thanks

**Language Note:** PeerJ staff have identified that the English language needs to be improved. When you prepare your next revision, please either (i) have a colleague who is proficient in English and familiar with the subject matter review your manuscript, or (ii) contact a professional editing service to review your manuscript. PeerJ can provide language editing services - you can contact us at copyediting@peerj.com for pricing (be sure to provide your manuscript number and title). – PeerJ Staff

Reviewer 1 ·

Basic reporting

This study presents a significant advancement in understanding the stress response mechanisms of Caenorhabditis elegans to landfill leachate, focusing on trehalose synthesis. The comprehensive experimental design, which includes the comparison of different landfill leachate treatment processes, provides valuable insights into the physiological and genetic adaptations of nematodes to environmental stressors. The findings on trehalose's role in enhancing nematode tolerance to toxic substances are intriguing and contribute to the broader field of environmental toxicology and stress biology.

However, there are areas where the manuscript could be strengthened:

Broader Contextualization: The discussion could further explore the implications of these findings in the context of landfill leachate management and potential biotechnological applications for environmental remediation.

Future Directions: The conclusion succinctly summarizes the findings but could also suggest specific avenues for future research, such as investigating the potential for engineering trehalose metabolic pathways in other organisms for bioremediation purposes.

Overall, the article contributes valuable knowledge to the field and, with some revisions, could have a significant impact on both the scientific community and environmental management practices.

Experimental design

Clarification of Methodologies: Some sections could benefit from more detailed explanations of the experimental procedures to ensure reproducibility. For instance, the description of the trehalose measurement techniques could be elaborated upon.

Validity of the findings

no comment

Reviewer 2 ·

Basic reporting

no commen

Experimental design

no comment

Validity of the findings

no comment

Additional comments

The manuscript titled "The nematode Caenorhabditis elegans enhances tolerance to landfill leachate stress by increasing trehalose synthesis" explores how C. elegans, a model organism, responds to the toxic effects of landfill leachate. It focuses on the nematode's increased synthesis of trehalose, a disaccharide known for its protective effects against stress, as a key mechanism for enhancing tolerance. The study evaluates the physiological toxicity of waste leachate tail water from different treatment processes on C. elegans, considering survival, growth, development, reproduction, and the expression of trehalose-related genes. The findings indicate that exposure to landfill leachate leads to decreased glucose and glycogen content and increased trehalose content, suggesting an adaptive response to adversity through trehalose accumulation. The study aims to provide insights into the stress response of nematodes exposed to toxic substances and develop a molecular model for evaluating waste leachate discharge effects.
The major remarks on the manuscript concern (for details, see below):
The abstract could be more concise and focused on the main findings of the study.
The introduction could provide more background information on the importance of studying the effects of landfill leachate on Caenorhabditis elegans.
The author provides a detailed description of the treatment methods used in the process of online insect exposure, but more relevant literature is needed to support the rationality of the experiment, especially in terms of the period and time of online insect exposure.
The discussion section could provide more insights into the implications of the study and its contribution to the field.
Detail comments:
Line 149: Correct “;” to “.”
Line 153-154: Correct the sentence: Body length, width of each nematode weas processed with Image J software (Jena, Germany).
Line 187-188: Please check this words: concerntration, analysic, and anneling.
Line 207: Formatting errors.
Line 210, 211, 212: the author used x g, is it standardized.
Line 270: Change “On the whole,” to “Overall”.
Line 279: Correct “trehalose related” to “trehalose-related”.
Line 313: Correct “et al” to “et al.”.
Table 1: Why add QCe before the gene name, the author should explain it in the caption or main text.
Figure 3: Although Figure 3 indicates the scale, more details should be added, such as the magnification of the microscope during shooting

---

## Round 0.2 · accepted · Accept

Congratulations!
Thanks for submitting your work to PeerJ.

Reviewer 1 ·

Basic reporting

The revised manuscript answered all my comments. I have no further comments.

Experimental design

The revised manuscript answered all my comments. I have no further comments.

Validity of the findings

The revised manuscript answered all my comments. I have no further comments.

Additional comments

The revised manuscript answered all my comments. I have no further comments.

Reviewer 2 ·

Basic reporting

no comment

Experimental design

no comment

Validity of the findings

no comment

Additional comments

no comment